# Non-stationary statistics and formation jitter in transient photon condensation

Benjamin T. Walker[1,2], João D. Rodrigues ID [1✉], Himadri S. Dhar[1], Rupert F. Oulton ID [1], Florian Mintert[1] & Robert A. Nyman ID [1]

While equilibrium phase transitions are easily described by order parameters and free-energy landscapes, for their non-stationary counterparts these quantities are usually ill-defined. Here, we probe transient non-equilibrium dynamics of an optically pumped, dye-filled microcavity. We quench the system to a far-from-equilibrium state and find delayed condensation close to a critical excitation energy, a transient equivalent of critical slowing down. Besides number fluctuations near the critical excitation energy, we show that transient phase transitions exhibit timing jitter in the condensate formation. This jitter is a manifestation of the randomness associated with spontaneous emission, showing that condensation is a stochastic, rather than deterministic process. Despite the non-equilibrium character of this phase transition, we construct an effective free-energy landscape that describes the formation jitter and allows, in principle, its generalization to a wider class of processes.

[1] Physics Department, Blackett Laboratory, Imperial College London, Prince Consort Road, London SW7 2AZ, UK. [2] Centre for Doctoral Training in Controlled Quantum Dynamics, Imperial College London, Prince Consort Road, London SW7 2AZ, UK. ✉email: j.marques-rodrigues@imperial.ac.uk

The connection between the properties of a system at thermal equilibrium and the geometry of its free-energy landscape is a powerful concept that dates back to original ideas developed by Gibbs[1]. This relation is particularly relevant near a second-order phase transition, where all the relevant macroscopic details are described by an emergent order parameter. This is, on average, located at the minimum of the free-energy landscape[2,3], with its neighbourhood being locally probed as the system is driven through configuration space by fluctuations (thermal or quantum). While we still lack a universal generalization of these ideas to non-equilibrium systems, Jaynes has suggested its possibility[4]. Some well-established, near-equilibrium stochastic descriptions of relaxation involving free-energy surfaces are known[5], but are not necessarily valid far from equilibrium. A particular example in this direction has been constructed for the laser[6], a fundamentally non-equilibrium system whose steady state can be described as the minimum of a properly defined effective free energy, corresponding to a detailed balance between driving and dissipation.

While the previous arguments are relevant for systems close to a steady state, a sudden parameter change, often called a quench, necessarily brings the system sufficiently far from equilibrium to question the validity of such approaches. The meaning of a quench depends on context and, in particular, one can distinguish between Hamiltonian and non-Hamiltonian cases. The former consist of time-dependent variations in some sort of interaction term, involved, for instance, in the Mott insulator-superfluid transition[7,8] or the build-up of anti-ferromagnetic correlations in Ising models[9]. Non-Hamiltonian quenches contain a more general class of processes. In cold atoms, for instance, the Kibble–Zurek mechanism[10,11] is observed by evaporatively cooling the system at a finite rate, quenching the system through a BEC phase transition. We shall refer to a quench as a sudden change in one of the system parameters that brings it to a far-from-equilibrium state, without affecting its Hamiltonian.

Photon condensates are ideal platforms to explore both equilibrium and non-equilibrium physics. A thermalizing medium, typically a dye solution, is placed inside an incoherently pumped optical microcavity. The combined rates of thermalization, pumping and cavity loss enable such a driven-dissipative system to be tuned between in- and out-of-equilibrium regimes[12–15]. Following the initial observation of Bose–Einstein condensation of photons[16], a number of experiments on grand-canonical fluctuations[17], spontaneous symmetry breaking[18], emergence of long-range order[19], among other aspects of equilibrium physics[20,21] have been described. Their non-equilibrium counterparts, however, remain greatly unexplored.

Here, we study the transient dynamics of photon condensation that follows a quench in a dye-filled optical microcavity. Besides measuring the ensemble-averaged photon number dynamics, we introduce the non-stationary, two-time, second-order correlation function $g^{(2)}(t_1, t_2)$. It provides access to the statistical properties of the photon condensation transition, and is particularly relevant in non-stationary systems, when the full knowledge of individual realizations is inaccessible. While the usual stationary correlation function, $g^{(2)}(\tau)$, accurately accounts for fluctuations in steady state, $g^{(2)}(t_1, t_2)$ is the appropriate quantity to describe the evolution of transient, non-equilibrium systems. The averaged condensate intensity as a function of time shows width broadening, a manifestation of diverging jitter in the condensate formation time upon approaching the critical excitation energy. This effect is directly witnessed by distinctive off-diagonal anti-correlations in $g^{(2)}(t_1, t_2)$ and originates from quantum fluctuations associated with spontaneous emission. By properly defining an effective (non-equilibrium) free energy, we suggest that jitter may be a universal feature of transient phase transitions in systems obeying relatively general conditions on the convexity of their free-energy landscape.

## Results

**Microscopic cavity model.** Despite the fundamentally multi-mode character of our optical cavity, the phenomenology described here is essentially that of a single-mode system. Cavity excitations (photons and excited molecules) can be lost by two processes: mirror transmission and molecular spontaneous emission into free space, at rates $\kappa$ and $\Gamma_\downarrow$, respectively. The essentials of the cavity dynamics are described by the density operator $\rho$, for both photons and molecules, which obeys the master equation[12,22,23]

$$\frac{d\rho}{dt} = -i[H_0, \rho] + \kappa \mathcal{L}[\hat{a}]\rho + \sum_{k=1}^{N_{\mathrm{mol}}} \Big\{ \Gamma_\downarrow\, \mathcal{L}[\sigma_k^-] + \Gamma_\uparrow\, \mathcal{L}[\sigma_k^+] + A\,\mathcal{L}[\hat{a}\,\sigma_k^+] + E\,\mathcal{L}[\hat{a}^\dagger\sigma_k^-] \Big\}\rho, \quad (1)$$

with the Hamiltonian for the bare cavity $H_0 = \hat{a}^\dagger\hat{a}$, $E$ and $A$ the dye emission and absorption rates, respectively, $\Gamma_\uparrow$ the incoherent (external) pumping rate and $N_{\mathrm{mol}}$ the total number of molecules inside the cavity. In general, $\Gamma_\uparrow$ is a time-dependent quantity, such as in the case of pulsed pumping. Due to the high collision rate between dye and solvent molecules, all the relevant cavity processes, including light–matter interactions, are incoherent.

Mean-field rate equations are obtained by taking expectation values and neglecting correlation terms in Eq. (1). The number of cavity photons $n = \langle \hat{a}^\dagger\hat{a} \rangle$ and fraction of excited molecules $f = \sum_k \langle \sigma_k^+ \sigma_k^- \rangle / N_{\mathrm{mol}}$, with $\langle\cdot\rangle$ denoting the (quantum mechanical) ensemble average, are then determined by

$$\frac{dn}{dt} = (E + A)(f - f_{\mathrm{c}})n + Ef, \quad \text{and} \quad (2)$$

$$\frac{df}{dt} = -\Gamma_\downarrow f + \frac{An}{N_{\mathrm{mol}}}(1 - f) - \frac{E(n+1)}{N_{\mathrm{mol}}}f + \Gamma_\uparrow(1 - f). \quad (3)$$

Here, the critical excitation fraction is defined as

$$f_{\mathrm{c}} = \frac{\kappa + A}{A + E}, \quad (4)$$

and, in the limit of high number of photons, corresponds to the transition point between a net increase or decrease in the number of cavity photons with time. At $f = f_{\mathrm{c}}$, and in the absence of pumping and losses ($\kappa = \Gamma_\downarrow = \Gamma_\uparrow = 0$), an equilibrium (steady state) between molecular excitations and photons is established by a principle of detailed balance[13]. The photon number in this equilibrium state would show a phase transition as the total number of cavity excitations, $N_{\mathrm{ex}} = n + fN_{\mathrm{mol}}$, which is the control parameter, is increased. The photon number, or order parameter, ranges from a disordered phase ($n \lesssim 1$) dominated by spontaneous emission to an ordered phase ($n \gg 1$) dominated by stimulated emission. While there is, in principle, a U(1) symmetry breaking upon crossing the condensation phase transition, the full dynamics can be described simply through photon number $n$[12]. By exciting a large number of dye molecules over a short period of time, the cavity can be quenched through this phase transition to a far-from-equilibrium state. The subsequent relaxation dynamics correspond to a non-stationary, transient counterpart of the equilibrium phase transition described above. In this way, we define a transient phase transition as the evolution in configuration space after a jump across a phase transition in parameter space (a quench), where "phase transition" has its usual time-independent, thermodynamic meaning. This is distinct from the recently introduced concept of dynamical phase transitions[24,25]. The non-linear coupling between photons and molecular excitations occurring

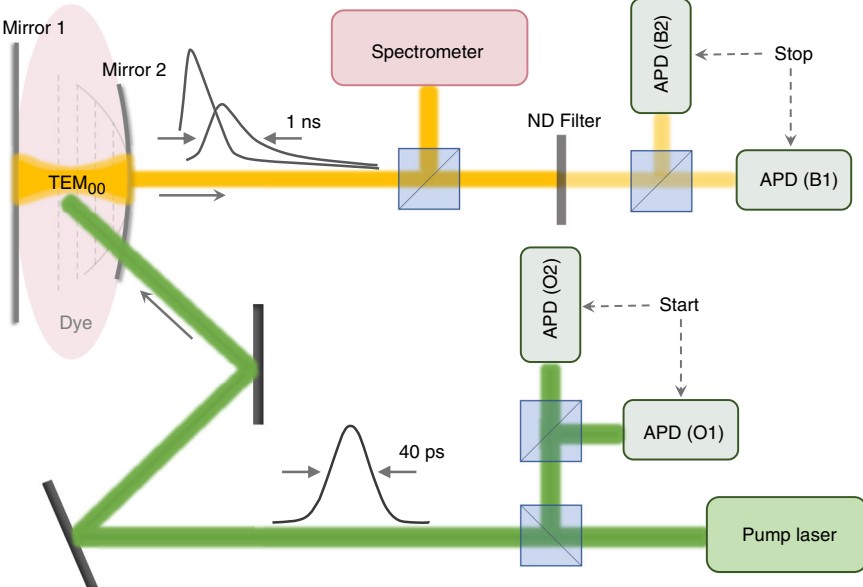

**Fig. 1 Schematic representation of the experimental setup.** The dye-filled microcavity is composed of one planar (1) and one spherical (2) mirror. The pulsed pump (green) is transmitted through the cavity at an angle of ~52°. The transverse ground-state mode (yellow) leaks through one of the mirrors and is directed both into a spectrometer and a pair of single-photon detectors, B1 and B2, in a Hanbury Brown–Twiss arrangement. A second pair of detectors, O1 and O2, is used to time the beginning of the experiment.

during this transient relaxation process gives rise to non-trivial fluctuation and correlation properties. Finally, given its lossy character, the light will transition back to the phase dominated by spontaneous emission before all excitations are lost.

A few notes are in order regarding the multi-mode nature of our cavity. Within the single-mode approximation, the rate term $\Gamma_\downarrow$ accounts for emission both into free space and cavity modes that do not reach the regime of stimulated emission (do not condense), which will be discussed in more detail later. Also, and despite not being relevant for the results discussed here, effects associated with the multi-mode character as well as spatially resolved molecular reservoirs have been appreciated in the context of gain clamping[26] and decondensation mechanisms[14].

**Experimental setup**. The experimental configuration is sketched in Fig. 1. The optical cavity is composed of one planar and one spherical mirror of 0.25 metres radius of curvature, which traps the photons. The cavity is filled with a 2 mM solution of rhodamine-6G in ethylene glycol. All the essential dynamics occur at the tenth longitudinal mode, corresponding to a cavity length of approximately 2 $\mu$m. A 40 ps laser pulse at 532 nm, typically ranging from 0.5 to 2 nJ in energy, is used to rapidly excite the molecules, quenching the cavity to a far-from-equilibrium state. In response, a much longer pulse ($\gtrsim$1 ns) of light leaks from the cavity mirrors, the exact temporal shape of which depends both on the cavity parameters (loss rate, dye concentration, emission and absorption rates) as well as the number of molecular excitations that follow the pump pulse, which is the control parameter used to select the different dynamical phases. A portion of pump light is directed onto two saturated avalanche single-photon detectors (APDs), O1 and O2, where a coincident detection is used as a time stamp for the beginning of the experiment, with a measured uncertainty of about 10 ps. The cavity output light is directed onto two unsaturated APDs, B1 and B2, with an average of 0.1 detections per pulse, on each detector. The experiment is conducted at a repetition rate of 11 kHz. Such a low repetition rate ensures a complete decay of all excitations and statistical independence between

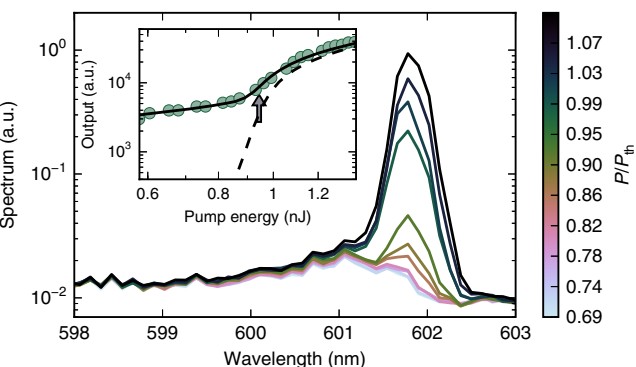

**Fig. 2 Spectrum of the cavity output.** The figure shows spectra above and below the condensation threshold at the critical excitation energy $P_{th}$. The spectral peak is located at the fundamental mode (ground state) of the cavity, or cavity cutoff, at approximately 602 nm. The inset depicts the total cavity output (dots) and comparison with the single-mode mean-field model, with (full line) and without (dashed line) the contribution from the spontaneous emission background. The arrow indicates the threshold point.

different realizations. We describe the experimental results in the form of following three sets:

(1) zero-time statistics: full time-averaged cavity output;
(2) one-time statistics: time-resolved, but averaged over all forms of fluctuations and correlations in the cavity output;
(3) two-time statistics: unequal time, cross-correlated signal from detectors B1 and B2, providing access to fluctuations in the cavity output.

**Zero-time statistics**. We begin by demonstrating the existence of a (condensation) phase transition in the total amount of light emitted by the cavity as the excitation energy, or pump energy, $P$ is increased beyond a critical value $P_{th}$, as shown in Fig. 2. This increase in the cavity output corresponds to the onset of stimulated emission, which rapidly de-excites the molecules. As such, a

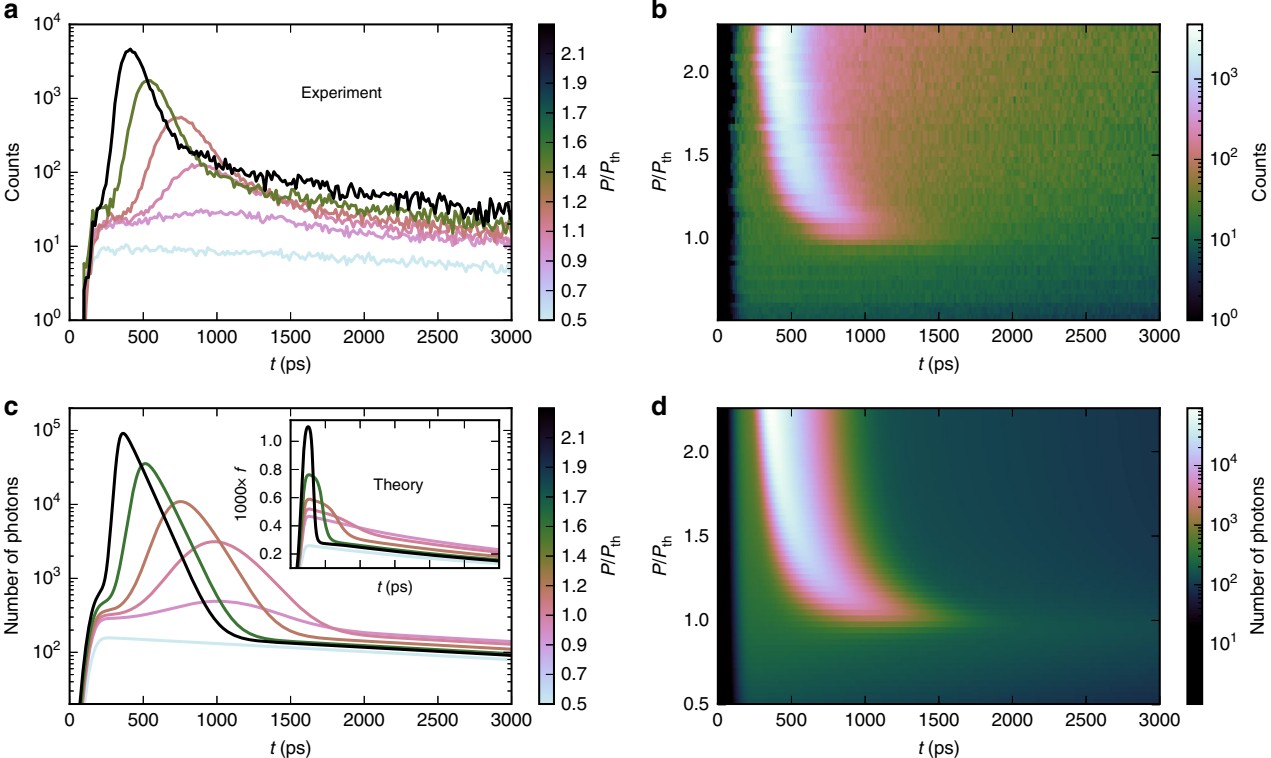

**Fig. 3 Time-dependent cavity output intensity.** The pump pulse sets the beginning of the time axis. We observe a delay in the growth of the photon condensate close to the critical excitation fraction $P_{th}$, accompanied by a large pulse broadening. Experimental results are shown in **a** and **b**, while simulations of the mean-field rate equations are shown in **c** and **d**. The inset depicts the experimentally inaccessible molecular excitation fraction $f$, during the same timespan as the main axes, demonstrating the two-way coupling between photons and molecules. The cavity ground state is located at 602 nm (cavity cutoff), the same conditions as in Fig. 2. Panels **a**, and **c** depict a subset of the data on **b** and **d**.

larger number of molecules decay into cavity modes rather than into free space, thus increasing the detected signal. The spectra also show a tendency towards thermalization, witnessed by robust condensation in the cavity ground state and indicating a regime where photon reabsorption plays a significant role[13,14]. Consequently, and despite the absence of a full thermal distribution, parallels may be drawn with Bose–Einstein condensation of photons[16,21,27,28]. While BEC is only strictly defined in thermal equilibrium as the macroscopic occupation of the ground state, we are assuming here a broader concept of condensation, as discussed in such diverse fields as physics, ecology, network theory or social sciences[28–31]. This can be thought of as the process where a particular, or small set of modes, in a multi-mode system becomes macroscopically occupied while the remaining ones saturate or become depleted.

By counting the rate of detection events in B1 and B2, we measure the total cavity output as a function of input pulse energy (input-output, or light-yield, curves), as shown in the inset of Fig. 2. The threshold, or critical, excitation energy can then be defined as the inflection in the light-yield curve. The signal from the cavity is collected without any filtering and multi-mode fibres are used to couple light into the detectors. Since non-condensed modes are also coupled to the APDs, we model their contribution by defining the detected signal $I_{det}$ as the sum of the single condensing cavity mode and a background of spontaneous emission, $I_{det} \propto \kappa n + \alpha \Gamma_{\downarrow} f N_{mol}$. Here, $\alpha$ is an empirical parameter set by fitting the light-yield curve to the mean-field rate equations. Loosely speaking, $\alpha$ determines the fraction of spontaneous emission into non-modelled cavity modes and is expected to be a small contribution, which will be verified in "One-time statistics" section.

**One-time statistics**. Here, we expand the time-averaged results of the previous section to the time-dependent cavity output pulse shape, as shown in Fig. 3. By collecting unlabelled detections on both B1 and B2, we effectively average over any form of correlations and fluctuations. Pulses that form below the threshold excitation energy ($P < P_{th}$) display a simple exponential decay on a time scale of about $\tau_0 \sim 4$ ns, the molecular excited-state lifetime. Above threshold, stimulated emission becomes important, leading to a large increase in photon number, followed by rapid depopulation of the condensate before a final decay at the slower time scale of the molecular excited-state decay.

It is instructive at this point to reflect upon the interplay and coupled dynamics of the molecular excitation fraction, $f$, and the number of cavity photons, $n$. In equilibrium, the molecular excitation fraction, $f$, cannot exceed its critical value, $f_c$. Under non-equilibrium conditions, however, if at any instant $f > f_c$ (e.g. after a quench), the photon population will grow exponentially until $f$ drops below $f_c$. This exponential increase in the number of photons, resulting from the onset of stimulated emission, is accompanied by rapid de-excitation of molecules, as shown in Fig. 3. Such a two-way coupling between photons and molecules is at the origin of the phenomenology described in this paper. It is worth noting that the rates of emission and absorption determine the relative size between the molecular excitation reservoir, $f N_{mol}$, and the photon number $n$. We choose experimental parameters that make the number of photons comparable with $f N_{mol}$, such that the effects of this two-way coupling become more prominent. On the opposite limit of $f N_{mol} \gg n$, which is approached for lower values of $\lambda_0$ (the cavity cutoff wavelength), the larger molecular reservoir becomes insensitive to photon number

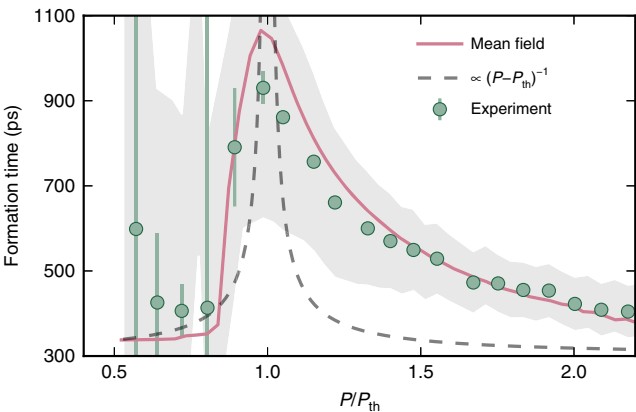

**Fig. 4 Condensate formation time.** The condensate formation time, defined as the interval between the pump pulse and the peak photon number, displays a transient analogue of critical slowing down. The formation time and corresponding error bars are obtained by fitting Gaussian profiles in a neighbouring region around the peak photon number. The full-red and black-dashed lines are the mean-field simulations and the critical (power-law) divergence of condensate formation time for the lossless case, respectively. The shaded area depicts the increasing pulse width upon approaching the critical excitation energy.

fluctuations, approaching the limit of a Markovian bath. This limit is at the origin of the observation of grand-canonical number statistics in photon BECs[17].

We fit $\Gamma_\downarrow$ and $\kappa$ to the results in Fig. 3 and find $\kappa = 10^{10}\,\mathrm{s}^{-1}$, corresponding to a cavity lifetime of 100 ps, and $\Gamma_\downarrow = 0.998\Gamma_0$, with $\Gamma_0 = 1/\tau_0$ the molecular fluorescence decay rate. Within the single-mode approximation, this means that only 0.2% of the total molecular emission goes into the condensing mode while 99.8% goes both into free space and excited modes that do not reach the regime of stimulated emission. Together with the light-yield curves in Fig. 2, the contribution of spontaneous emission coupling into the detectors is found to be $\alpha = 0.13$; small as expected. The emission and absorption rates are not taken as fitting parameters but rather calculated from experimental absorption and emission data for rhodamine-6G[32]. The total number of molecules is calculated from the dye concentration and cavity volume to be $N_{\mathrm{mol}} = 1.9 \times 10^8$.

As $f$ approaches $f_c$ from above, the cavity dynamics become slow, as dictated by Eq. (2). In particular, since $\dot{n} \propto f - f_c$, one might expect critical slowing down in the condensate formation time, with a critical exponent of $-1$, assuming $\dot{f} \sim 0$[33]. However, the presence of direct spontaneous emission into free space prevents $f$ from remaining close to $f_c$ for long times and the entire relaxation process is necessarily transient. Despite the mechanism of true critical slowing down being frustrated, we still observe a slowing in the time taken for the condensate to form as we approach the critical excitation energy from above, as shown in Fig. 4. By comparing to the critical divergence for the lossless case, where formation time is proportional to $(P - P_{\mathrm{th}})^{-1}$, we observe a broadening of the threshold region due to the lossy nature of the cavity. Besides this transient analogue of critical slowing down, a distinct feature emerges upon approaching the critical excitation energy, which contributes to the broadening of the average output pulse. In the next section, we show that this originates from a particular form of fluctuations that arise in such transient phase transitions: jitter in the condensate formation time.

**Two-time statistics.** Correlations and fluctuations of the cavity output can now be investigated by retaining the labelling of

detection timestamps in B1 and B2. We then construct the two-time, non-stationary, second-order correlation function $g^{(2)}(t_1, t_2)$. Second-order correlations are typically described by the single-time $g^{(2)}(\tau)$ function, with $\tau = t_1 - t_2$, due to time-translation symmetry in steady-state conditions. In transient systems, however, the absence of this symmetry means that the full two-time, $t_1$ and $t_2$, dependence must be retained. We can then define

$$g^{(2)}(t_1, t_2) = \frac{\langle a^\dagger(t_1) a^\dagger(t_2) a(t_2) a(t_1) \rangle}{\langle a^\dagger(t_1) a(t_1) \rangle \langle a^\dagger(t_2) a(t_2) \rangle} \approx \frac{P(t_1, t_2)}{P(t_1) P(t_2)}, \quad (5)$$

where $P(t_1, t_2)$ is the joint probability of photon detection at times $t_1$ and $t_2$ in detectors B1 and B2, respectively. By marginalizing over the second detector, $P(t_1)$ and $P(t_2)$ are obtained as the single-detector probabilities. The approximation in Eq. (5) is accurate as long as $[a^\dagger(t_1), a(t_2)] \approx 0$ or $\langle a^\dagger(t) a(t) \rangle \gg 1$. The former is satisfied when $|t_1 - t_2|$ is larger than the coherence time (much smaller than all relevant time scales involved in the cavity dynamics), and the latter is true for large photon numbers, as verified in Fig. 3.

The second-order correlation function is shown in Fig. 5. The two main features to be retained here are the diagonal positive correlation ($g^{(2)} > 1$) and the off-diagonal anti-correlation ($g^{(2)} < 1$) lobes. These features are mainly a manifestation of the same kind of fluctuations—jitter, or shot-to-shot timing fluctuations, in the condensate formation—which become amplified near the critical excitation energy. In the remainder of this section, we discuss this effect associated with transient phase transitions.

Let us proceed by separately analysing diagonal and anti-diagonal correlations, as shown in Fig. 6. For equal times, $g^{(2)}$ provides immediate information on number, or intensity, fluctuations, namely $g^{(2)}(t, t) \simeq 1 + \langle \Delta n(t)^2 \rangle / \langle n(t) \rangle^2$. As such, periods of larger fluctuations coincide with the inflection point of the average pulse shape, consistent with a condensate forming at slightly different instants in each realization of the experiment. In a microscopic picture of the cavity dynamics, spontaneously emitted photons are required to seed the condensate growth. The randomness associated with the quantum nature of spontaneous emission then leads to such shot-to-shot time fluctuations, or jitter in the condensate formation. As we shall demonstrate in the next section, these periods of larger fluctuations correspond to a passage through the convex part of an effective free-energy landscape.

The above interpretation is further supported by the off-diagonal anti-correlation lobes, as seen in Figs. 5 and 6. Given the finite duration of the condensate pulse, if a photon is detected at an early time, it is less likely that another photon will be detected at a later time. In other words, the whole light pulse is either early or late. Off-diagonal regions with $g^{(2)} < 1$ are then an immediate witness of fluctuations in formation time. Also, it is further evidence of the two-way coupling between photons and molecules, as discussed previously.

By retaining correlations up to second-order in Eq. (1) using a cluster expansion (or higher-order cumulants)[34–37], the two-time correlation function can be obtained via the quantum regression theorem[38], as shown in Fig. 5. Details of this approach can be found in "Methods" section. An alternative method to capture correlations to all orders is to construct a quantum trajectories (or Monte Carlo wavefunctions) approach[39–42]. Different classes of events (molecular emission and absorption, cavity loss, etc.) are defined and drawn at random given their respective rates. These are dynamically calculated as the number of photons and molecular excitations are updated at each step. Full details of this model can be found in "Methods" section. In the limit of a large number of realizations, this approach is equivalent to

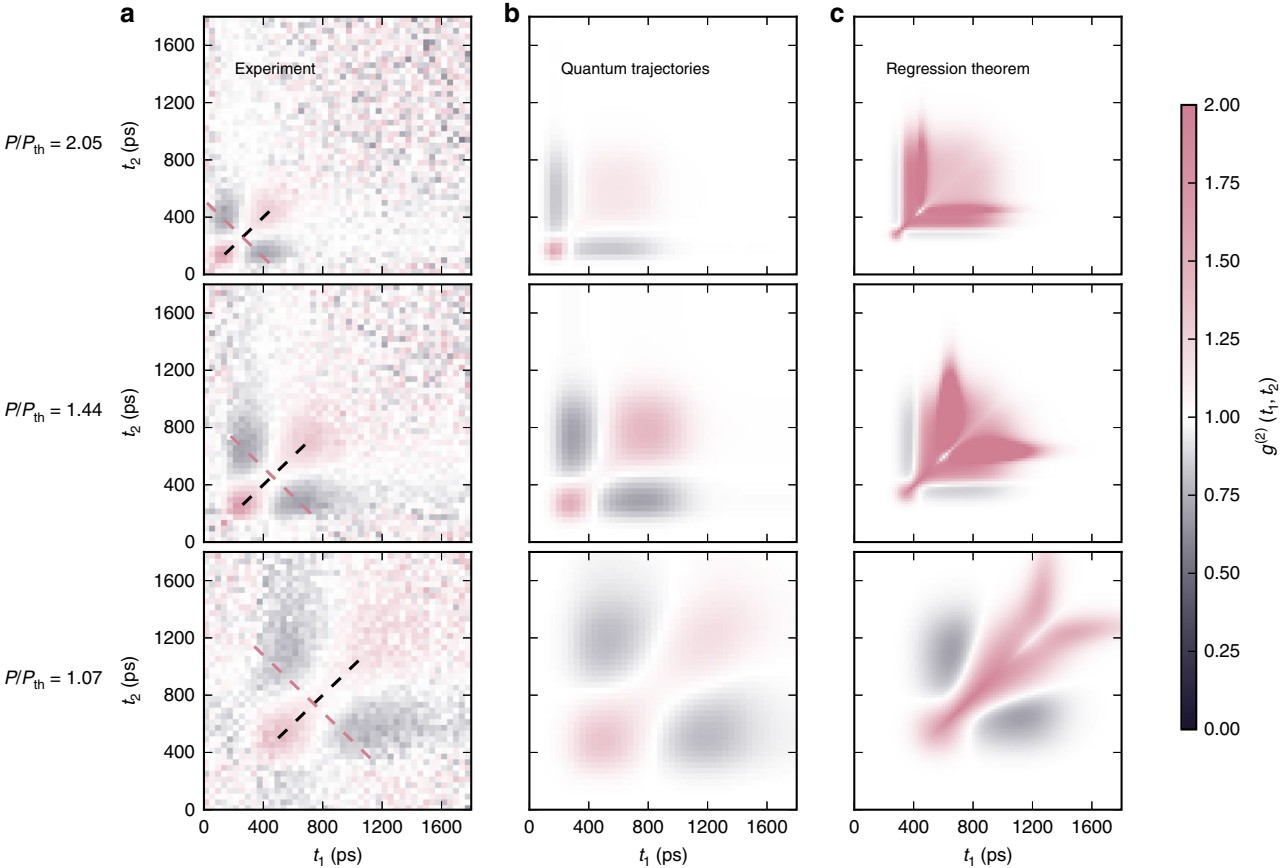

**Fig. 5 Two-time, non-stationary, second-order correlation function for various excitation energies.** Experimental data (**a**) can be compared with quantum trajectories simulations (**b**) and a semi-analytic quantum regression approach based on a master equation expansion up to second order (**c**). In both cases, good agreement with the experiment requires a spontaneous emission background, as discussed in "Zero-time statistics" section. The dashed curves in the experimental data correspond to the diagonal and off-diagonal regions of $g^{(2)}(t_1, t_2)$ depicted in greater detail in Fig. 6. The cavity cutoff is set to $\lambda_0 = 595$ nm. Despite the slightly lower cutoff wavelength than used in "One-time statistics" section, the qualitative features of the average cavity output are the same as before.

evolving the density matrix according to Eq. (1). By retaining correlations to all orders, the quantum trajectories show the best match to data. However, the quantum regression approach could be modified to include higher-order correlations. This may be an interesting approach since the quantum regression, given its semi-analytical character, allows a faster sampling of different parameters.

The experiment does not allow direct access to individual trajectories, only the effect of their relative fluctuations on $g^{(2)}$. However, the quantum trajectories method allows us to easily appreciate the effect of formation jitter, depicted in Fig. 7. The good agreement between experiment and theory in Figs. 5 and 6 is evidence that individual trajectories in the experiment have a similar form to those depicted in Fig. 7. Here, the formation jitter becomes clear, with larger shot-to-shot fluctuations occurring close to the critical excitation energy, which contributes to the pulse broadening described in "One-time statistics" section. The trajectories are shown to diverge only in the very early stages of the experiment, where the number of photons in the condensing mode is low and spontaneous emission is the dominant cavity process, the latter thus being at the origin of the stochastic nature of condensation and its associated formation jitter. The exact form of $g^{(2)}(t_1, t_2)$ depends on both the individual pulse shapes and their uncertainty in formation time. As it turns out, the earlier forming pulses (relatively far above threshold) are of shorter duration than later forming pulses (close to the critical point). This effect competes with the larger fluctuations in

formation time closer to threshold, such that a diverging behaviour may not be extractable from the $g^{(2)}$ maps alone.

Note that the fluctuations described here are of a different origin than those arising from the grand-canonical nature of a photon BEC[17], which predicts $g^{(2)}(0) = 2$. In the latter, a steady state is achieved by a detailed balance between cavity loss and continuous pumping, with the fluctuations being related to the coupling between the photons and the molecular grand-canonical reservoir in conditions of thermal equilibrium. There is time-translation symmetry and number fluctuations are damped within a 2 ns time scale[17]. In contrast, the $g^{(2)}$ structure we identify here reflects the propagation of the initial fluctuations associated with the spontaneous emission events that trigger the growth of the condensate pulse. The system never reaches a steady state and all the dynamics are fundamentally transient. In principle, $g^{(2)}(t, t)$, which depends on both the individual pulse shapes and their formation jitter, can even be larger than 2.

**Effective free energy.** We now develop a general treatment of the relaxation process described in "Results" section, thus highlighting its universal features and applicability outside the particular case of photon condensates. Relaxation of an order parameter $\psi(t)$ towards its equilibrium value $\psi_0$ can be generically modelled by the time-dependent Landau equation[5,43–45]

$$\frac{d\psi}{dt} = -\frac{\partial F}{\partial \psi} + \eta(t), \qquad (6)$$

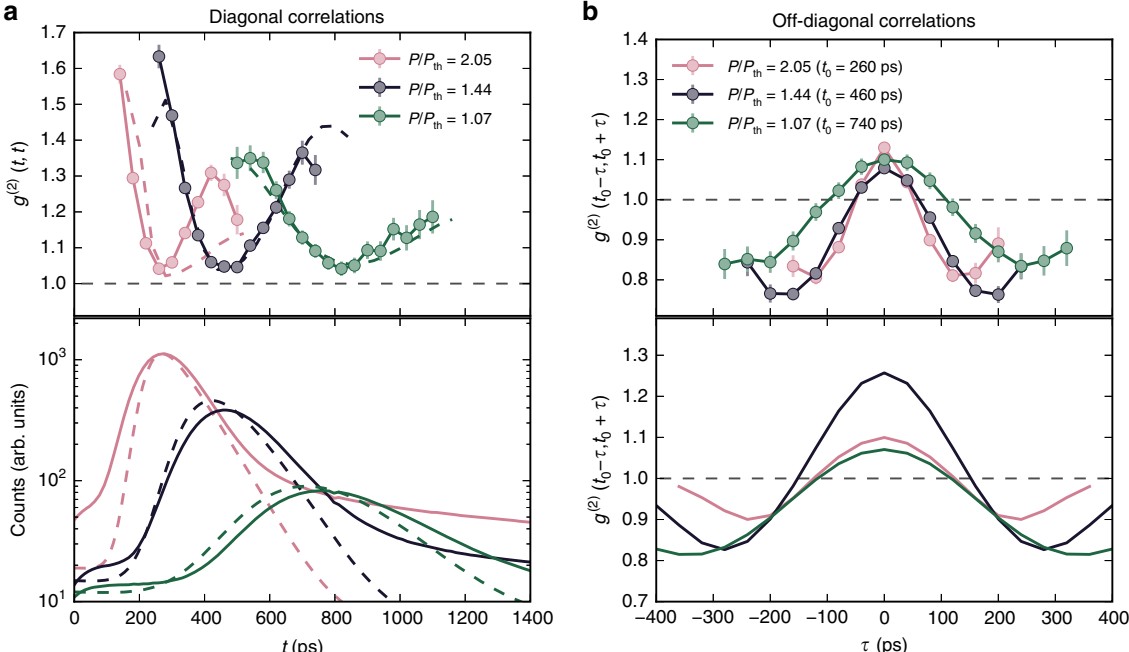

**Fig. 6 Diagonal and off-diagonal correlations.** Cuts through the regions depicted by the dashed lines in Fig. 5. **a** Diagonal correlations, $g^{(2)}(t, t)$, shown on top, and the average condensate pulse, shown on the bottom. The full and dashed lines depict the experimental data and the quantum trajectories results, respectively. **b** Off-diagonal correlations, $g^{(2)}(t_0 - \tau, t_0 + \tau)$, with $t_0$ the peak time of the average condensate pulse. Top depicts the experimental results, with the quantum trajectories simulation on the bottom. Here, the theory slightly deviate from the experimental results, which may be attributed to the error that propagates from determining the peak time $t_0$.

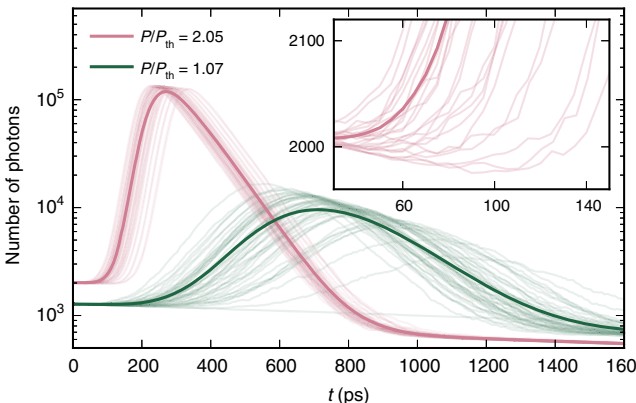

**Fig. 7 Quantum trajectories simulation.** Fifty individual trajectories are shown in light colours, while darker colours depict their (ensemble) average. The number of photons includes those from the single condensing mode as well as the spontaneous emission background discussed in "Zero-time statistics". The parameters match those of Figs. 5 and 6. The inset depicts a zoomed view of the early stages of condensation.

with $F = F(\psi)$ the near-equilibrium free energy and $\eta$ a generic Langevin stochastic force. This defines a universal class of dissipative relaxation processes[5], typically valid near thermal equilibrium, where $F = F(\psi)$ can be expanded in Taylor serious around $\psi_0$. We shall undertake here a different approach that will extend the validity of the model above into far-from-equilibrium conditions.

In the absence of cavity losses and pumping ($\kappa = \Gamma_\downarrow = \Gamma_\uparrow \equiv 0$), the full non-equilibrium dynamics described by Eqs. (2) and (3) can be formally mapped onto the time-dependent Landau equation by integrating the dissipative term of Eq. (6), which

defines the effective free energy[6,46,47]

$$
\mathrm{F}(n) = -\int_0^n \frac{dn'}{dt} \, dn' = -\frac{E N_{\mathrm{ex}}}{N_{\mathrm{mol}}} n - \left[ \frac{E(N_{\mathrm{ex}} - 1)}{N_{\mathrm{mol}}} - A\left(1 - \frac{N_{\mathrm{ex}}}{N_{\mathrm{mol}}}\right) \right] \frac{n^2}{2}
$$
$$
+ \frac{(E + A)}{N_{\mathrm{mol}}} \frac{n^3}{3},
\tag{7}
$$

where the number of photons becomes the order parameter ($\psi \rightarrow n$) and total number of excitations, $N_{\mathrm{ex}} = n + f N_{\mathrm{mol}}$, the control parameter, as defined earlier. By retaining the full dynamical information contained in the mean-field rate Eqs. (2) and (3), we effectively map the dynamics of photon condensates onto the geometrical properties of the effective free-energy landscape defined by Eq. (7). Notably, equilibrium free energies are only valid in the vicinity of the equilibrium point, yet the effective free-energy landscape in Eq. (7) goes beyond this limitation and correctly describes the relaxation of the photon condensate when prepared in any non-equilibrium configuration in the closed-system approximation. We thus bring the relaxation of both near-equilibrium and far-from-equilibrium systems into a similar mathematical framework, in the spirit first anticipated by Jaynes[4]. Figure 8 depicts the effective free energy for different (initial) excitation fractions, which allows for a direct analogy with the experiment, where the pump energy determines the initial excitation fraction, with the cavity being initialized with $n = 0$. Moreover, all the parameters used here match those of "Results" section. Interestingly, despite being extended into the non-equilibrium regimes, the effective free-energy landscape defined by Eq. (7) and depicted in Fig. 8 shares the same geometrical features as those defined for near-equilibrium conditions[48], providing evidence for universal properties regarding the relaxation of both near- and far-from-equilibrium systems.

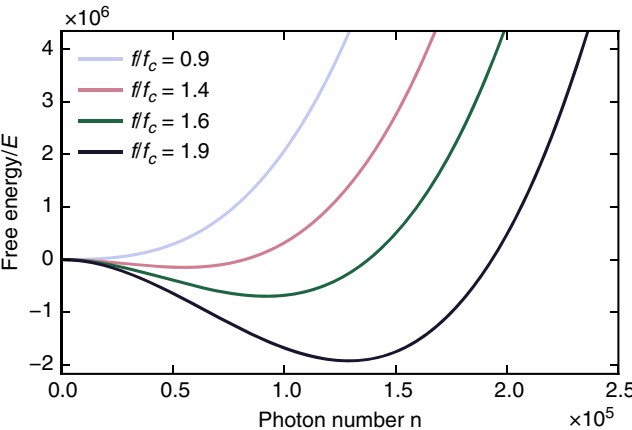

**Fig. 8 Effective free-energy landscapes.** Free energy for the microcavity photon number (order parameter), calculated for different values of the initial molecular excitation fraction $f$. Besides being the quantity directly varied in the experiment, $f$ determines the control parameter as $N_{ex} = n + f N_{mol}$.

In the photon condensate context, the stochastic force $\eta$ accounts for spontaneous emission, such that $\langle \eta(t)\eta(0) \rangle = f^2 E^2 \delta(t)$[49,50]. This allows us to include fluctuations and beyond-mean field effects in an universal model of relaxation. An alternative, and formally equivalent, approach would be the construction of a Fokker–Planck equation[5] for the probability density function (PDF) $P_n(t)$, with a drift term given by the derivative of the effective free energy, as in Eq. (6), and a diffusion term describing the fluctuations from spontaneous emission. This PDF encodes qualitative information about the fluctuation properties of the order parameter as it transiently evolves from the far-from-equilibrium state that follows a quench. We construct such a PDF by considering a large set of random walks (50,000) evolving according to Eq. (6) over the free-energy landscape defined in Eq. (7)—Fig. 9.

Quite generally, fluctuations act to broaden the photon number PDF while a positive (concave) curvature in the free energy tends to localise it. At a second-order phase transition, the curvature at the minimum of the free energy (defining an equilibrium order parameter) vanishes and the PDF shows diverging fluctuations that persist for long times, giving rise to critical slowing down. In transient, non-equilibrium systems dramatic features also occur, with regions of negative (convex) curvature acting to amplify fluctuations. A PDF evolving through these regions while relaxing towards the free-energy minimum experiences a short-lived but large increase of fluctuations, as shown in Fig. 9. The jitter described in "Results" section is the immediate consequence of this. The maximum of number fluctuations, marked by the peak in $g^{(2)}(t, t)$, and shown in Fig. 6, occurs when the average order parameter reaches the free-energy inflection point. From Eq. (6), this corresponds to the (temporal) inflection point of the order parameter, in complete agreement with the results depicted in the previous section. The width of the photon number PDF then shrinks back as the order parameter evolves to the concave part of $F$. While the free-energy landscape is more convex for larger values of the control parameter, which increases number fluctuations, these are longer-lived close to threshold (larger jitter). This effect witnesses the complex interplay between number and time fluctuations associated with the stochastic transient relaxation process determined by Eq. (6). Even in the approximation of a closed system, the free-energy model correctly predicts the major features observed in the experiment, namely the slowing down of the condensate formation accompanied by increasing timing jitter and its relation with photon number fluctuations.

In the presence of loss by cavity transmission, the free energy and photon number PDF are coupled in a non-trivial way. For sufficiently large $\kappa$, as in the case of the experiment described in "Results" section, the rate of change of the free-energy landscape depends on the photon number $n$, which is itself described by a given PDF. The landscape is now neither constant, nor a simple function of time, but rather coupled to the photon number history, such that for trajectories where the condensate forms early, it also decays early, leading to the anti-correlation lobes seen in $g^{(2)}(t_1, t_2)$. This is essentially the same result as depicted by the quantum trajectories simulation in Fig. 7 but reinterpreted under the geometrical properties of the effective free-energy landscape.

The free-energy description assumes the total number of cavity excitations $N_{ex}$, the control parameter, to be fixed, such that all molecular excitations are converted into cavity photons, corresponding to the limit of negligible losses. As a final remark, Eq. (6) allows for a formal reconstruction of the free-energy landscape, $F(\psi)$, from the observed average dynamics of the order parameter, $\psi(t)$, although in practice the results are not very informative.

## Discussion

In this work, we have described the transient non-equilibrium dynamics of light in a dye-filled optical cavity quenched through a condensation phase transition. By rapidly exciting a large number of dye molecules, the system is brought to a far-from-equilibrium state. By averaging over all forms of fluctuations, we observed a delayed formation of the condensed phase, interpreted as a transient equivalent of critical slowing down. When quenched above the condensation threshold excitation energy, the quantum fluctuations associated with spontaneous emission seed the growth of the order parameter as the system relaxes into equilibrium. The relaxation dynamics is slower close to the critical point, a feature easily interpreted under the geometrical properties of the effective free-energy landscape, which becomes flat. The same mechanism is responsible for the usual critical slowing down in the relaxation rate of the ordered phase that follows a second-order phase transition. Also, despite the absence of latent heat and the fact that we are dealing with second-order and not first-order phase transitions, analogies can be drawn with the precipitation in supercooled, or supersaturated, liquids. Even quenched above the critical point, a seed of spontaneously emitted photons is needed to nucleate condensation, playing the role of the seeding crystals in supercooled, or supersaturated, liquids. Also, once seeded, crystallization across the entire liquid is faster for liquids quenched further across their critical parameters, with temperature playing the same role as the excitation fraction that follows the quench, in the optical cavity context.

By measuring the statistical properties of this transient condensation, we describe a novel form of diverging fluctuations around the critical point, jitter in the formation of the ordered phase. These are witnessed by strong diagonal correlations and off-diagonal anti-correlations in the non-stationary, second-order correlation function, $g^{(2)}(t_1, t_2)$. More precisely, we demonstrated that while the diagonal of $g^{(2)}$ is a powerful probe of the geometrical properties of the free-energy landscape, its off-diagonal elements reflect the relevant dissipation processes, with the anti-correlation lobes a joint effect of jitter and cavity loss. Fluctuations, arising from spontaneous emission, are highly amplified as the order parameter goes through the convex part of the free-energy landscape towards its equilibrium point.

The description in terms of the geometric properties of the effective free-energy landscape, being independent of the microscopical details of our particular system, allows us to generalize our observations. In particular, both the transient critical slowing down and the jitter in the formation of the order parameter are

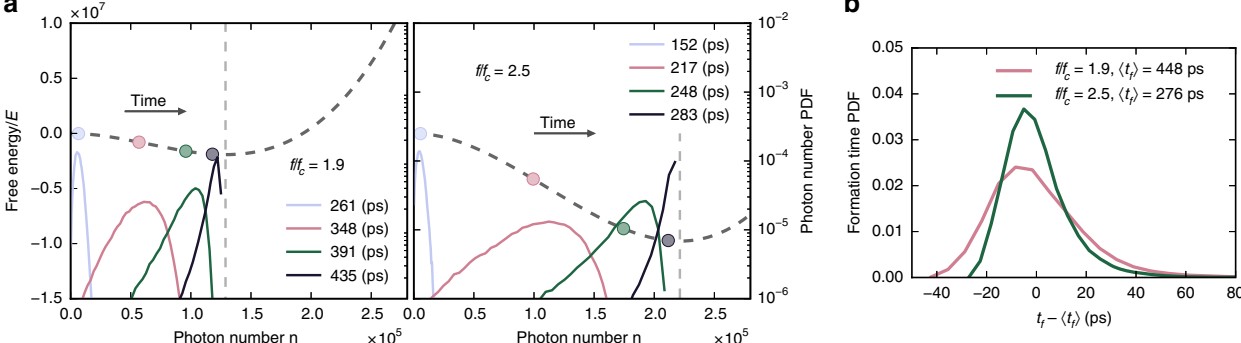

**Fig. 9 Free-energy description of non-equilibrium photon condensation. a** The photon number probability distribution function (PDF) (coloured lines), describing fluctuations of the order parameter for two different initial excitation fractions. The circles indicate the expected value of $n$ at a given time instant. While the latter is essentially Gaussian noise, the free-energy convexity far from the equilibrium point induces non-Gaussian, heavy-tailed statistics in photon number. This skewed statistics partially explains the limitations of the quantum regression model described in "Results" section. **b** The PDF of the condensate formation time, taken as the instant at which the photon number of each realization grows beyond 80% of that which minimizes the free energy (equilibrium point). We observe both the slowing down of the condensate formation, as well as the increasing jitter closer to threshold.

expected to be universal features of the dynamics that follows a quench through a second-order phase transition. In micro- and nano-lasers, in particular, the full two-time, non-stationary analysis of the relaxation process has been greatly overlooked and previous results[51–54] may now benefit from being re-examined. Despite some recent efforts in describing time fluctuations and other non-equilibrium features of micro- and nano-lasers[55–57], and to the best of our knowledge, we present here for the first time a generic and comprehensive description of the relation between temporal and number fluctuations in the non-stationary dynamics of systems undergoing second-order phase transitions. Finally, the system studied in this work, as well as the related examples stated above can be described by single-value order parameters. One may wonder on the generalization of these effects in spatially extended systems, where the order parameter is a function of both space and time. In the context of the Kibble–Zurek mechanism[10,11], for instance, most studies are simply concerned with the defect number scaling after the system relaxes to some steady state, with the intrinsic relaxation dynamics often ignored. As such, although we cannot anticipate specific effects, one wonders about the correspondence between transient fluctuation dynamics of the zero-dimensional system described in our paper and that of spatially extended systems.

## Methods

**Second-order rate equations and the quantum regression theorem.** From the non-equilibrium model introduced in Eq. (1), one can derive rate equations for the ensemble-averaged photon number, $\langle n \rangle = \langle \hat{a}^\dagger(t)\hat{a}(t) \rangle$, and the number of excited molecules, $\langle m \rangle = \sum_k \langle \sigma_k^+(t)\sigma_k^-(t) \rangle$, as

$$\frac{d\langle n \rangle}{dt} = -\kappa\langle n \rangle + E\{\langle m \rangle + \langle nm \rangle\} A\{\langle n \rangle N_{mol} - \langle nm \rangle\}, \quad (8)$$

$$\frac{d\langle m \rangle}{dt} = -E\{\langle m \rangle + \langle nm \rangle\} + A\{\langle n \rangle N_{mol} - \langle nm \rangle\} - \Gamma_\downarrow\langle m \rangle + \Gamma_\uparrow(N_{mol} - \langle m \rangle). \quad (9)$$

The calculation of $\langle nm \rangle$ depends on the estimation of $\langle n^2 m \rangle$, $\langle nm^2 \rangle$, $\langle n^3 \rangle$, ... , which requires solving a large number of ordinary differential equations. These can be reduced with an hierarchical set of approximations. For instance, in the semi-classical limit, the expectation values for $n$ and $m$ are factorized, $\langle mn \rangle \approx \langle m \rangle\langle n \rangle$, reducing Eq. (9) to

$$\frac{dn}{dt} = -\kappa n + Ef(n+1) - An(1-f), \quad (10)$$

$$\frac{df}{dt} = -\Gamma_\downarrow f + An(1-f) - E(n+1)f + \Gamma_\uparrow(1-f). \quad (11)$$

These are equivalent to Eqs. (2) and (3). Here, we define $f = \langle m \rangle/N_{mol}$ as the molecular excitation fraction and set, for the ease of notation, $n = \langle n \rangle$. Despite

ignoring correlations all together, this corresponds to a first-level approximation to the non-equilibrium cavity dynamics.

In order to account for correlations and fluctuations, one needs to go beyond the semi-classical approximation. In particular, the expectation values can be expanded in a hierarchical manner[34–37] given by

$$\sigma_{xy}^2 = \langle xy \rangle - \langle x \rangle\langle y \rangle, \quad (12)$$

$$\sigma_{xyz}^3 = \langle xyz \rangle - \sum \sigma_{xy}^2\langle z \rangle - \langle x \rangle\langle y \rangle\langle z \rangle, \quad (13)$$

$$\sigma_{wxyz}^4 = \langle wxyz \rangle - \sum \sigma_{wxy}^3\langle z \rangle - \sum \sigma_{wx}^2\langle y \rangle\langle z \rangle - \sum \sigma_{wx}^2\sigma_{yz}^2 - \langle w \rangle\langle x \rangle\langle y \rangle\langle z \rangle. \quad (14)$$

These represent the second, third, and fourth order cumulants, with the summation referring to all possible combination of variables. A minimal description of correlations is constructed by truncating the hierarchy at second order. In this way, and by defining $\sigma_x^2 = \langle x^2 \rangle - \langle x \rangle^2$, with $x = \{n, m\}$, we explicitly write

$$\frac{dn}{dt} = -\kappa n + E\{(n+1)m + \sigma_{nm}^2\} - A\{n(N_{mol} - m) - \sigma_{nm}^2\}, \quad (15)$$

$$\frac{dm}{dt} = -\Gamma_\downarrow m - E\{(n+1)m + \sigma_{nm}^2\} + \Gamma_\uparrow(N_{mol} - m) + A\{n(N_{mol} - m) - \sigma_{nm}^2\}, \quad (16)$$

$$\frac{d\sigma_n^2}{dt} = -\kappa(n + 2\sigma_n^2) + E\{(n+1)m + 2\sigma_n^2 m + \sigma_{nm}^2(2n+1)\} - A\{n(N_{mol} - m) + 2\sigma_n^2(N_{mol} - m) - \sigma_{nm}^2(2n-1)\}, \quad (17)$$

$$\frac{d\sigma_m^2}{dt} = -\Gamma_\downarrow(m + 2\sigma_m^2) - E\{-(n+1)m + 2\sigma_m^2(n+1) + \sigma_{nm}^2(2m-1)\} + A\{n(N_{mol} - m) - 2\sigma_m^2 n + \sigma_{nm}^2(-2m + 2N_{mol} - 1)\} + \Gamma_\uparrow(N_{mol} - m - 2\sigma_m^2), \quad (18)$$

$$\frac{d\sigma_{nm}^2}{dt} = -(\kappa + \Gamma_\downarrow + \Gamma_\uparrow)\sigma_{nm}^2 + E\{(n+1)(-m + \sigma_m^2) - \sigma_n^2 m + \sigma_{nm}^2(m - n - 2)\} + A\{-n(N_{mol} - m) + \sigma_m^2 n + \sigma_n^2(N_{mol} - m) + \sigma_{nm}^2(m - n + 1 - N_{mol})\}. \quad (19)$$

The second-order photon correlation function at zero-time delay, $g^{(2)}(t)$, follow immediately as

$$g^{(2)}(t) = \frac{\langle \hat{a}^\dagger(t)\hat{a}^\dagger(t)\hat{a}(t)\hat{a}(t) \rangle}{\langle \hat{a}^\dagger(t)\hat{a}(t) \rangle^2} = \frac{\langle n^2(t) \rangle - \langle n(t) \rangle}{\langle n(t) \rangle^2} = 1 + \frac{\sigma_n^2(t) - n(t)}{n^2(t)}. \quad (20)$$

The two-time second-order correlation function can be obtained by invoking the quantum regression theorem[38], which allows us to calculate any quantity of the form $\langle X(t+\tau)Y(t) \rangle$ using two single-time evolutions. Let the initial state of the system be $\chi(0)$, and the evolution be given by the map, $\chi(t) = \mathcal{V}(t, t')\chi(t')$. The two-

time expectation value can then be written as

$$\langle X(t+\tau)Y(t)\rangle = \mathrm{Tr}[X\mathcal{V}(t+\tau,t)\{Y\chi(t)\}] = \mathrm{Tr}[X\mathcal{V}(t+\tau,t)\{Y\mathcal{V}(t,0)\chi(0)\}]. \quad (21)$$

The two-time function is thus calculated by evolving $\chi(0)$ from 0 to $t$, followed by the conditional state $Y\chi(t)$ from $t$ to $t+\tau$. For our cavity model, we begin by first evolving the density operator, $\rho$, from $t=0$ to $t=t_1$, using the second-order rate Eqs. (15) and (19), obtaining $g^{(2)}(t_1)$. Second, the first-order rate Eqs. (2) and (3) are used to evolve the conditional state, $\tilde{\rho} = \hat{a}(t_1)\rho\hat{a}^\dagger(t_1)/\langle\hat{a}^\dagger(t_1)\hat{a}(t_1)\rangle$ from $t=t_1$ to $t=t_2$. Following Eq. (21), one then arrives at the two-time photon correlation function

$$g^{(2)}(t_1,t_2) = \frac{\langle\hat{a}^\dagger(t_1)\hat{a}^\dagger(t_2)\hat{a}(t_2)\hat{a}(t_1)\rangle}{\langle\hat{a}^\dagger(t_1)\hat{a}(t_1)\rangle\langle\hat{a}^\dagger(t_2)\hat{a}(t_2)\rangle}. \quad (22)$$

**Quantum trajectories approach**. The second-order approach described above corresponds to a first-level approximation to the description of correlations and fluctuations in the cavity dynamics. Moving to higher-order expansions increases the number of ordinary differential equations needed to resolve the dynamics, which soon becomes cumbersome and impractical. An alternative approach to solve the master Eq. (1) is to use the quantum trajectories (or Monte Carlo wavefunction) method[39–42]. Here, the Lindblad dynamics of the density operator $\rho$ is replaced by a wavefunction whose evolution is given by a non-Hermitian effective Hamiltonian, interspersed with stochastic quantum jumps. Subsequently, evolution of $\rho$ is approximated by an ensemble average of wavefunctions, or trajectories, say $|\psi_i\rangle$. For a large number of trajectories, $z$, the average of any observable is then given by

$$\langle\hat{X}(t)\rangle = \mathrm{Tr}[\hat{X}\rho(t)] \approx \frac{1}{z}\sum_{i=1}^{z}\langle\psi_i(t)|\hat{X}|\psi_i(t)\rangle. \quad (23)$$

The effective non-Hermitian Hamiltonian for the non-equilibrium cavity model in Eq. (1) is given by

$$\mathcal{H}_{\mathrm{eff}} = H_0 - \frac{i}{2}\sum_k J_k^\dagger J_k, \quad (24)$$

where $J_k$ are the jump operators defining the stochastic dynamics. In the non-equilibrium cavity model, coherences cannot be created by $\mathcal{H}_{\mathrm{eff}}$. Hence, if a quantum trajectory starts in a particular number state, say $|\psi_i(0)\rangle = |n_0, m_0\rangle$, the action of $\mathcal{H}_{\mathrm{eff}}$ alone does not change the state in this number basis. The complete dynamics of the trajectory is simply governed by the stochastic jumps $J_k$, occuring at rates $R_k$:

$$\sqrt{\kappa}\,\hat{a} : |n,m\rangle \rightarrow |n-1,m\rangle; R_0 = \kappa n, \quad (25)$$

$$\sqrt{\Gamma_\uparrow}\,\sigma^+ ; |n,m\rangle \rightarrow |n,m+1\rangle; R_1 = \Gamma_\uparrow(N-m), \quad (26)$$

$$\sqrt{\Gamma_\downarrow}\,\sigma^- : |n,m\rangle \rightarrow |n,m-1\rangle; R_2 = \Gamma_\downarrow n, \quad (27)$$

$$\sqrt{E}\,\hat{a}^\dagger\sigma^- : |n,m\rangle \rightarrow |n+1,m-1\rangle; R_3 = E(n+1)m, \quad (28)$$

$$\sqrt{A}\,\hat{a}\sigma^+ : |n,m\rangle \rightarrow |n-1,m+1\rangle; R_4 = An(N-m). \quad (29)$$

A particular quantum trajectory is constructed by drawing a series of stochastic events, with their individual probabilities proportional to the rates $R_k$. The time between consecutive events is drawn from an exponential distribution, whose mean is the inverse of total rate of events. From a large ensemble of trajectories, we can calculate the non-stationary second-order correlation function, $g^{(2)}(t_1, t_2)$, as

$$g^{(2)}(t_1,t_2) = \frac{\langle n(t_1)n(t_2)\rangle}{\langle n(t_1)\rangle\langle n(t_2)\rangle}, \quad (30)$$

where $\langle\cdot\rangle$ denotes ensemble average, over the entire set of trajectories. The same approximations as discussed in "Two-time statistics" section are assumed here as well.

## Data availability

The data related to this paper may be requested from the authors or via https://dataenquiryEXSS@imperial.ac.uk.

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

## Acknowledgements

We acknowledge financial support from EPSRC (UK) through the grants EP/S000755/1, EP/J017027/1, the Centre for Doctoral Training in Controlled Quantum Dynamics EP/L016524/1 and the European Commission via the PhoQuS project (H2020-FETFLAG-2018-03) number 820392. We also thank Julian Schmitt for helpful discussions.

## Author contributions

B.T.W. and J.D.R. carried out the experiments and analysed the data, with assistance from R.A.N. and R.F.O; H.S.D., J.D.R. and B.T.W. worked out the theory and ran the simulations, with assistance from F.M.; J.D.R. and B.T.W wrote the manuscript with input from all authors.

## Competing interests

The authors declare no competing interests.
