## [Peer Review File · Nature Communications]

Reviewers' comments:

Reviewer #1 (Remarks to the Author):

The paper by Walker et al studies the transient dynamics of formation and decay of a photon Bose-Einstein condensate. The paper contains a good mix of experimental and theoretical results with the experiments in good agreement with the theoretical predictions. The paper is technically sound and the results are of interest to a specialised audience but in its present form I do not clearly see why the paper meets the necessary criteria required by Nature Communications.

To me the discussion in the introduction is not so clearly connected to the rest of the paper. The main part contains a few very general statements about how free energy landscapes can be connected to equilibrium phase transitions and generalisations of this to non-equilibrium systems. These ideas are not new, they are just being applied to the photon BEC for the first time. The discussion then jumps very abruptly to a brief description of the system that the authors consider. It would be better if there was some introduction to the types of physics which have been observed in other photon BEC experiments to help set the context for these results.

I also have some more technical clarifications which the authors should address before resubmitting the manuscript:

In the introduction the phrase 'the previous arguments are relevant for systems close to a steady state'. It is important to note that these type of free energy landscapes can only in general be constructed for the steady state of systems where mean-field theory is good. Going beyond this is a challenging task see e.g. arXiv:1910.04777 for recent developments in this direction.

How is the pulse implemented in the numerical simulations? I guess Γ_{\uparrow} is time dependent?

In Figure 3 it might be useful to show what the theoretical prediction of the molecular excitation fraction is doing through the time evolution shown.

Why do the trajectory and regression calculations give such different behaviour? Naively the trajectories should give identical results to a master equation treatment. Is it because the regression calculation is really only at the level of rate equations? Is it possible to quantify when these two approaches significantly differ?

The stochastic unravelling of a particular master equation is not unique and which one to use depends on the particular experimental setup. From what I can tell the authors use the quantum jump unravelling. Is it clear that this is the correct interpretation for their experiments?

The authors mention the distinction between the molecular excitations acting as a Markovian or non-Markovian reservoir in different parameter regimes. Is it possible to derive a Markovian equation for just the photons and then compare this to the full results to show in a more quantitative way that this is the case?

It is not clear why the formal order parameter for the phase transition should be the number of photons? Usually for BEC and lasing transitions it is the phase of the macroscopic wavefunction which is the order parameter. As far as I understand the model here has a $U(1)$ symmetry which the photon number does not detect a spontaneous breaking of?

Reviewer #2 (Remarks to the Author):

The manuscript describes an investigation of the condensate formation in a dye-filled optical microcavity. The system is pumped with a pulsed laser, and the authors monitor (besides the temporal evolution) the photon statistics upon the formation of the photon condensate. A detailed comparison with theory calculation is performed, and the results can be described by an effective free-energy description with convexity far from equilibrium points resulting in increased fluctuations. Comparison to theory also shows that the observed jitter in condensate formation stems from spontaneous emission. To my judgement, some revisions of the manuscript are required:

-the authors should more clearly describe why there is evidence for the spontaneous photons being responsible for the jitter. Also, they should more clearly say why there is evidence for non-Markovian dynamics. These issues are key points. Also in some other parts of the manuscript, a clearer description of the physics results would help.

-I do not see how the results can have implications to the growth of collided nanoparticles (last sentence in abstract). In addition, I think the connection to condensates in diverse other field as "physics, ecology, network theory or social sciences" (pages 3) is misleading.

-eq. 6: Is this a universal definition of an "effective free energy" or just the definition used in Ref. 5? I am not a theorist, but my feeling is that the authors here should be more precise.

-end of page 8: The term "non-Markovian" means that the bath has a memory. This should not have anything to do with whether the emission goes into free space or cavity modes.

-Discussion (page 9): I assume Refs [43-45] were works much further away from thermal equilibrium, and associated to laser dynamics? The authors probably should mention this.

-page 5: I assume 0.2 percent is the fraction of the total emission that couples into the ground state mode (not that of all cavity modes)? Perhaps the photons that have decayed to higher cavity modes are no more relevant in this experiment because the spontaneous decay time is longer than the complete time of the experiment, given that simulated emission only occurs for the ground mode. At least a brief discussion on these issues would help, otherwise the above number is confusing.

- Fig. 2 (inset): Why does the signal increase upon condensation? Is this because the detector predominantly monitors the ground state mode (the collection efficiency for the other modes is smaller), or because in the absence of stimulated emission the molecules remain in the upper state for so long that in the detected time window the molecules have not undergone significant decay?

-Caption of Fig. 9: The authors should also say what the difference between the two plots is and explain it better.

Reviewer #3 (Remarks to the Author):

The manuscript by Walker et al. presents a study of photon condensation under pulsed excitation. The experimental results are presented together with a theoretical model, showing good agreement. The subsequent analysis puts a strong emphasis on nonequilibrium free energy.

The experimental results seem to be robust and the theoretical modeling is done at various levels of sophistication, including a quantum trajectory description (also used in W. Verstraelen and M. Wouters, PRA 100, 013804 (2019)). As explained well in the manuscript, the theoretical description based on time evolution of the correlators truncated at second order performs less well. They find that the time at which the condensate forms fluctuates from shot to shot, because

it is determined by fluctuations. As a consequence, the $g_2(t_1, t_2)$ has a specific structure as a function of the time difference.

The main reservations I have with the manuscript concerns the interpretation in terms of nonequilibrium free energy. In the introduction, a reference to the discussion by Jaynes is given (Ref. [2]), but this work is quite general and nontechnical (the claim that a "bubble" of probability evolves in time is only made in the conclusions without much justification). Ref. [2] is once more cited in the discussion section, but I could not make a precise connection with the condensation jitter.

The further references seem to be mainly about equilibrium free energy (Refs. [3,4]), except for the laser (Ref. [5]). Therefore, it does not seem that the general framework behind the interpretation that is given to experiments seems not very solid.

I was actually a bit puzzled by the definition of the nonequilibrium free energy in Eq. (7) (I also didn't understand the integral in the first line). Before introducing it, driving and losses are set to zero, hence they are considering the closed system and the usual statistical mechanics definition of the free energy can be used: $\beta F = -\ln(p(n)) = \ln(N_{\text{ex}}! N_{\text{ground}}!) + \text{const}$, where "!" is factorial and $p(n)$ is the microcanonical probability to have n photons for a given total number of excitations $X=n+N_{\text{ex}}$ in the system. Perhaps there is a relation to Eq. (7), but I did not see it immediately by using Stirling's approximation.

Therefore, while I think that the experimental work and theoretical modeling with quantum trajectories is definitely a very valuable contribution to the study of photon condensates, I am not convinced by the robustness of the analysis in terms of the nonequilibrium free energy. Without framing the work in a broader context however, it lacks perhaps some appeal for the broad audience targeted by Nature Communications and is better suited for a more specialized journal.

My further questions/remarks are:

1) It is argued in the last paragraph of Sec. IIA that the emission into excited states can be treated as emission into free space. I am a bit skeptical toward this claim, because photons that are emitted into excited states can be reabsorbed, but the ones that are emitted into free space cannot. At the phenomenological level, this may be a reasonable first approximation though.

2) In the end of the first paragraph of Sec. IIC. it is said that the concept of BEC is not defined out of equilibrium. I think that this should be no problem, because the Penrose-Onsager criterion for Bose-Einstein condensation does not depend on thermal equilibrium, just on the one-body density matrix. I think that a more severe problem is the fact that the system is effectively zero-dimensional and therefore the thermodynamic limit which is needed for the Penrose-Onsager criterion cannot be easily taken (see also point 5).

3) The values of g_2 never approach the grand-canonical value of 2 in Fig. 6. Why is this the case?

4) In the introduction (first line of second page), the concept "transient phase transition" is put forward. It is not clear what is meant with this. Do the authors not refer to the transient dynamics after quenching through a phase transition instead of a "transient phase transition", which seems to indicate a different type of phase transition.

5) In the conclusions, they make the connection to 'phase transitions' in different systems. This may be a bit misleading since the system under consideration does not have a phase transition in a precise sense. Because only one mode is considered (it is argued that the excited states are not important) it is actually rather a zero-dimensional system, that does not show a phase transition: all quantities are analytic functions of parameters, except in the limit where the number of photons

goes to infinity. The final outlook of the paper is actually about what happens in spatially extended systems. It is actually well known that spatially extended systems that undergo sudden condensation feature Kibble-Zurek type vortices (cf. Ref. [9,10]). Maybe the authors could be more precise about the open problems in spatially extended systems.

Reply to Referees

Title: Nonstationary Statistics and Formation Jitter in Transient Photon Condensation

Authors: João D. Rodrigues, and co-authors, Ben Walker, Himadri Dhar, Rupert Oulton, Florian Mintert and Rob Nyman

We thank all three Referees for their well-considered replies. They largely agree on two key points: our experimental and theoretical work is carefully done, valid and interesting in its own right; but, the connection to the free-energy description is not quite clear enough to fully engage a broad audience. We start this reply by responding to all three Referees on this latter issue, before addressing the detailed points made by the Referees.

In our new version of the manuscript we have improved the connection between the dynamics of our system and the free-energy description by both deepening the discussion of generic relaxation models in terms of free energies (through extra references and discussion) and clarifying how we generalise the mapping from microscopic models to effective free-energy landscapes, so that predictions in the context of other systems can be performed.

We also agree that drawing on the concept of probability bubbles evolving in free-energy or entropy surfaces [1] is not sufficient to address the connection between any observed dynamics in our model and a more general free-energy landscape. In our modified manuscript, we have now shown that a much stronger connection can be made from the set of statistical equations of motion, such as the Langevin equations, as shown in Refs [2–5] for systems close to equilibrium. At this point, we simply map our well-established equations of motion [6], which are valid even far from equilibrium, to an effective free-energy landscape that clearly explains the key effect in our study, the condensation jitter. These observations are valid for all physical systems where the equations of motion map onto an equivalent free-energy landscape. Moreover, we show that our far-from-equilibrium free energy shares the same pertinent geometrical properties as those defined for near-equilibrium conditions. Therefore, the free-energy landscape we describe provides a powerful tool for predicting and explaining interesting phenomena in a large set of physical systems.

We have made four significant changes to the manuscript to reply to the Referees’ main points:

1. We have rewritten the first two paragraphs, as well as paragraph 4, of the Introduction to make a stronger link between established results in non-equilibrium physics and our ideas.
2. As suggested by Referee 1, we have now included a new paragraph introducing the physics of photon BECs.
3. We have rewritten the first 3 paragraphs of the “*Effective Free-Energy*” section to formalise our free-energy approach.
4. We have modified the “*One-time statistics*” section (mostly paragraph 2) to explain better the role of reservoir size on the coupling between photons and molecules.

We have also made other smaller changes, which we describe below in the detailed replies to the Referees’ remarks.

I. REPLY TO REFEREE 1

Referee 1 agrees that our work is “technically sound” but requests clarification of the general interest relating to our free-energy description, as we have explained above. Our reply to the more specific comments is:

Comment 1: “To me the discussion in the introduction is not so clearly connected to the rest of the paper. The main part contains a few very general statements about how free energy landscapes can be connected to equilibrium phase transitions and generalisations of this to non-equilibrium systems. These ideas are not new, they are just being applied to the photon BEC for the first time.

Reply: As stated in the beginning of this document in greater detail, we have re-written the introduction as to better connect our results with previous literature on non-equilibrium physics. A significant new aspect of our work is the application to far-from-equilibrium situations, where previous work mostly focused on close to equilibrium physics.

Comment 2: “The discussion then jumps very abruptly to a brief description of the system that the authors consider. It would be better if there was some introduction to the types of physics which have been observed in other photon BEC experiments to help set the context for these results.”

Reply: We accept this comment and have included a new paragraph in the introduction (the third paragraph).

Comment 3: “In the introduction the phrase “the previous arguments are relevant for systems close to a steady state”. It is important to note that these type of free energy landscapes can only in general be constructed for the steady state of systems where mean-field theory is good. Going beyond this is a challenging task see e.g. arXiv:1910.04777 for recent developments in this direction.”

Reply: We agree with the Referee and thank him/her for the suggested reference. We believe the discussion presented in the beginning of this document, together with the corresponding changes to the manuscript clarify not only the correctness of our free energy approach but also its validity to describe beyond mean-field effects.

Comment 4: “How is the pulse implemented in the numerical simulations? I guess Γ_{\uparrow} is time dependent?”

Reply: The Referee is correct. We have made this point clear after the statement of the master equation, Eq. (1).

Comment 5: “In Figure 3 it might be useful to show what the theoretical prediction of the molecular excitation fraction is doing through the time evolution shown.”

Reply: We have edited Fig 3 to include the evolution of the molecular excitation fraction.

Comment 6: “Why do the trajectory and regression calculations give such different behaviour? Naively the trajectories should give identical results to a master equation treatment. Is it because the regression calculation is really only at the level of rate equations? Is it possible to quantify when these two approaches significantly differ?”

Reply: As correctly mentioned by the Referee, the difference arises because the quantum regression calculations are based on a rate equation where operator correlations only up to the second order are retained. We have modified the text (Sec IIE, 5th paragraph) to explain that: *“By retaining correlations to all orders, the quantum trajectories show the best match to data. However, the quantum regression approach could be modified to include higher-order correlations. This may be an interesting approach since the quantum regression, given its semi-analytical character, allows a faster sampling of different parameters.”*

In principle, the quantum regression should give identical results to the quantum trajectories if significantly higher-order correlations are included in the rate equations. However, including even the third- or fourth-order correlations significantly reduces the efficiency of the quantum regression approach as compared to the quantum trajectories we employ here, without adding anything to the analysis.

Comment 7: “The stochastic unravelling of a particular master equation is not unique and which one to use depends on the particular experimental setup. From what I can tell the authors use the quantum jump unravelling. Is it clear that this is the correct interpretation for their experiments?”

Reply: All the experimental measurements (apart from the spectrum depicted in Fig. 2) rely on time-resolved single-photon detection. As such, quantum trajectories using quantum jumps to unravel the master equation would be the most appropriate for our experiments as compared to unravellings based on continuous stochastic processes, such as quantum state diffusion, which are more relevant for balanced homodyne or heterodyne detection based experiments. To keep the manuscript simple, we have chosen not to comment on this in the manuscript, but would include a sentence to this effect if the Referee requires it.

Comment 8: “The authors mention the distinction between the molecular excitations acting as a Markovian or non-Markovian reservoir in different parameter regimes. Is it possible to derive a Markovian equation for just the photons and then compare this to the full results to show in a more quantitative way that this is the case?”

Reply: In principle, one may construct a model where the photons are subject to a Markovian molecular bath. This is however outside of the scope of the current work. More importantly, as we mention in the manuscript, the complex interplay between time and number fluctuations we describe here is fully rooted in this non-Markovian character of the molecular bath or, equivalently, on the two-way coupling between photons and molecules. After a careful exploration of the parameter space in our theoretical model, we find that this non-Markovian regime is rather robust and, only in a narrow range of parameters does one actually approach the Markovian limit. All the results depicted here are deeply in the non-Markovian regime. As such, and to simplify the discussion on this subject, we have slightly rewritten both the second and the beginning of the third paragraphs of Sec IID. Other parts of the manuscript have also been slightly simplified, namely the last part of the free-energy section.

Comment 9: “It is not clear why the formal order parameter for the phase transition should be the number of photons? Usually for BEC and lasing transitions it is the phase of the macroscopic wavefunction which is the order parameter. As far as I understand the model here has a $U(1)$ symmetry which the photon number does not detect a spontaneous breaking of?”

Reply: The model does present a $U(1)$ symmetry. However, it is not our intent to describe this symmetry breaking upon condensation. This has been described elsewhere [7] by monitoring phase of the electric field E of the light emitted from the cavity. Given its symmetry, and in the absence of external symmetry-breaking fields, the model can be expanded in only even powers of E , namely $|E|^2$, $|E|^4$, etc. Apart from numerical factors, this is equivalent to an expansion in terms of powers of the photon number, n , n^2 , n^3 , etc, as defined in Eq. (7). There is no loss of generality in our treatment. We have added a sentence to this effect: *“While there is, in principle, a $U(1)$ symmetry breaking upon crossing the condensation phase transition, the full dynamics can be described simply through photon number n [6].”*

II. REPLY TO REFEREE 2

We appreciate the Referee’s detailed report. Below we reply to all comments:

Comment 1: “The authors should more clearly describe why there is evidence for the spontaneous photons being responsible for the jitter. Also, they should more clearly say why there is evidence for non-Markovian dynamics. These issues are key points. Also in some other parts of the manuscript, a clearer description of the physics results would help.”

Reply: The theoretical model accurately describes the experiment from the combined effects of essentially four stochastic processes: spontaneous (SpE) and stimulated (StE) emission, absorption (A) and cavity loss (L). The rates of StE, A and L are directly proportional to the photon number n , thus dominating the cavity dynamics as soon as the condensate begins to grow. On the other hand, SpE is the only relevant process in the very beginning of the experiment. We have modified Fig. (7), which now depicts a zoomed view of the trajectories diverging at early times, where spontaneous emission is the only relevant process. We added to the last paragraph of Sec IIE, upon the interpretation of Fig. (7), the following sentence: *“The trajectories are shown to diverge only in the very early stages of the experiment, where the number of photons in the condensing mode is low and spontaneous emission is the dominant cavity process, the latter thus being at the origin of the stochastic nature of condensation and its associated formation jitter.”*

Regarding the evidence for non-Markovian dynamics, we have rewritten the second paragraph of Sec IID in order to make this point more clear. Also, the inclusion of the molecular excitation fraction in Fig. (3), as suggested by Referee 1, clearly demonstrates this two-way coupling between photons and molecules. For reference, the quantum trajectories can be used to provide evidence of this intricate coupling in a different way - see Fig. (1) below.

FIG. 1: Demonstration of the two-way coupling between photons and molecules.

Comment 2: “I do not see how the results can have implications to the growth of collided nanoparticles (last sentence in abstract). In addition, I think the connection to condensates in diverse other field as “physics, ecology, network theory or social sciences” (pages 3) is misleading.”

Reply: The links to other fields are well-established in the general literature on statistical mechanics, e.g [8] (as cited in the manuscript) which explains how “condensation” is a very broadly applicable concept. We have added a sentence in the last paragraph (Discussion section) to better explain how our work can be applied to these other fields. Furthermore, we include citations justifying the use of a free-energy picture specifically in the case of colloidal

nanoparticle growth [9, 10], with a new sentence in the discussion (last paragraph) further describing this connection.

Comment 3: “Eq. 6: Is this a universal definition of an effective free energy or just the definition used in Ref. 5? I am not a theorist, but my feeling is that the authors here should be more precise.”

Reply: Eq. 6 defines a universal model of relaxation, for which we now cite several important papers in the manuscript. We believe the new discussion on the effective-free energy, described in the beginning of this document together with the corresponding list of changes to the manuscript and new citations achieves the precision the Referee is requesting.

Comment 4: “End of page 8: The term “non-Markovian” means that the bath has a memory. This should not have anything to do with whether the emission goes into free space or cavity modes.”

Reply: Besides referring back to the reply to comment 1, we invite the Referee to read paragraph 2 of the “*One-time Statistics*” section, where we discuss this manner in greater detail in the new version of the manuscript.

Comment 5: “Discussion (page 9): I assume Refs [43-45] were works much further away from thermal equilibrium, and associated to laser dynamics? The authors probably should mention this.”

Reply: Refs [43-45] (in the original first draft) are indeed related with distinct aspects of out-of-equilibrium laser dynamics. We clarify their context by modifying the sentence to: *“Despite some recent efforts in describing time fluctuations and other non-equilibrium features of micro- and nano-lasers [11–13], and to the best of our knowledge, we present here for the first time a generic and comprehensive description of the relation between temporal and number fluctuations in the non-stationary dynamics of systems undergoing second-order phase transitions.”*

Comment 6: “Page 5: I assume 0.2 percent is the fraction of the total emission that couples into the ground state mode (not that of all cavity modes)? Perhaps the photons that have decayed to higher cavity modes are no more relevant in this experiment because the spontaneous decay time is longer than the complete time of the experiment, given that simulated emission only occurs for the ground mode. At least a brief discussion on these issues would help, otherwise the above number is confusing.”

Reply: Within the single-mode approximation, one needs only to distinguish two decaying channels: decay into the condensing mode and into other channels, which include decay into free space and non-condensing excited cavity modes. We conclude that only 0.2 percent of the total molecular emission goes into the mode that condenses (or modes, though treated as one in the single-mode approximation). Consequently, 99.8 percent of molecular emission goes into both free space and cavity modes that don’t condense which, as the Referee states, occurs at the nanosecond timescale of the molecular excited state lifetime. As it turns out, the single-mode approximation suffices in describing the phenomenology we investigate here. We have changed the sentence “...meaning that only 0.2 percent of the total molecular emission couples to the cavity...” to: *“Within the single-mode approximation, this means that only 0.2 percent of the total molecular emission goes into the condensing mode while 99.8 percent goes both into free space and excited cavity modes that do not reach the regime of stimulated emission.”*

Comment 7: “Fig. 2 (inset): Why does the signal increase upon condensation? Is this because the detector predominantly monitors the ground state mode (the collection efficiency for the other modes is smaller), or because in the absence of stimulated emission the molecules remain in the upper state for so long that in the detected time window the molecules have not undergone significant decay?”

Reply: The sigmoid shape in Fig (2) is a classical manifestation of a phase transition. We use multimode fibres to couple the cavity emission into the detectors. Despite the coupling efficient being slightly mode-dependent, the signal increase upon crossing the phase transition threshold is related, as the Referee states, with the onset of stimulated emission which rapidly de-excites the molecules. As the Referee mentions, spontaneous emission happens on a longer timescale than that detected in the experiment. Accordingly, in the first paragraph of Sec (II C), we now include: *“This increase in the cavity output corresponds to the onset of stimulated emission which rapidly de-excites the molecules. As such, a larger number of molecules decay into cavity modes rather than into free space, thus increasing the detected signal”*

Comment 8: “Caption of Fig. 9: The authors should also say what the difference between the two plots is and explain it better.”

Reply: We changed the beginning of the caption to: *“Probability distribution function (PDF) of photon number (coloured lines), describing fluctuations of the order parameter for two different initial excitation fractions.”* In the end of the caption we also included *“While closer to threshold (top case), time fluctuations are larger (larger formation jitter), further above the critical excitation fraction (bottom case), number fluctuations are larger but contained to a narrower time interval. This witnesses the complex interplay between number and time fluctuations*

associated with the stochastic transient relaxation process determined by Eq. (6).”

III. REPLY TO REFEREE 3

We thank the Referee for the detailed analysis of our manuscript. The Referee inquiries about the use of the free-energy concept. We believe the discussion in the beginning of this document, together with the described changes to the manuscript, offers a clear reply to this issue. Below, we reply to the individual comments:

Comment 1: “The main reservations I have with the manuscript concerns the interpretation in terms of nonequilibrium free energy. In the introduction, a reference to the discussion by Jaynes is given (Ref. [2]), but this work is quite general and nontechnical (the claim that a “bubble” of probability evolves in time is only made in the conclusions without much justification). Ref. [2] is once more cited in the discussion section, but I could not make a precise connection with the condensation jitter. The further references seem to be mainly about equilibrium free energy (Refs. [3,4]), except for the laser (Ref. [5]). Therefore, it does not seem that the general framework behind the interpretation that is given to experiments seems not very solid.”

Reply: We hope the discussion enclosed in the beginning of this document (with the formal definition of a model of relaxation, and citations to pertinent literature) fully convinces the Referee about the correctness of our effective free-energy interpretation. Also, while Jaynes’ ideas are not placed in formal statement, our effective free-energy approach goes exactly in the direction anticipated in Ref [1]. As such, while this description correctly predicts and witnesses the universal character of the jitter effect, we acknowledge that the connection with the ideas put forward by Jaynes [1] is not immediate. We removed this claim from the Discussion section.

Comment 2: “I was actually a bit puzzled by the definition of the nonequilibrium free energy in Eq. (7) (I also didn’t understand the integral in the first line). Before introducing it, driving and losses are set to zero, hence they are considering the closed system and the usual statistical mechanics definition of the free energy can be used: $\beta F = -\ln(p(n)) = \ln(N_{ex}!N_{ground}!) + const$, where ! is factorial and $p(n)$ is the microcanonical probability to have n photons for a given total number of excitations $X = n + N_{ex}$ in the system. Perhaps there is a relation to Eq. (7), but I did not see it immediately by using Stirling’s approximation.”

Reply: The new discussion of the effective free-energy should clarify this issue. Also, the integral in the first line of Eq. (7) is merely a formal integration of Eq. (6), in the mean field limit, i.e. setting $\eta = 0$. Fluctuations are again recovered when computing the evolution of the photon number PDF.

Comment 3: “It is argued in the last paragraph of Sec. IIA that the emission into excited states can be treated as emission into free space. I am a bit skeptical toward this claim, because photons that are emitted into excited states can be reabsorbed, but the ones that are emitted into free space cannot. At the phenomenological level, this may be a reasonable first approximation though.”

Reply: In the single-mode approximation there are only two relevant decaying channels: decay into the mode that eventually condenses (reaches stimulated emission) and into every other channel, including decay into free space as well as into excited cavity modes that don’t reach the regime of stimulated emission. The Referee is right in the sense that photons in the latter case are still present in the cavity and can therefore be absorbed and re-emitted. As it turns out, however, the effects we describe here are not dependent on this distinction and, at the level of theoretical modelling, one can still phenomenologically account for both using the single Γ_{\downarrow} parameter.

Comment 4: “In the end of the first paragraph of Sec. IIC. it is said that the concept of BEC is not defined out of equilibrium. I think that this should be no problem, because the Penrose-Onsager criterion for Bose-Einstein condensation does not depend on thermal equilibrium, just on the one-body density matrix. I think that a more severe problem is the fact that the system is effectively zero-dimensional and therefore the thermodynamic limit which is needed for the Penrose-Onsager criterion cannot be easily taken (see also point 5).”

Reply: Our system is effectively zero-dimensional in the sense that it is well-described by a single, scalar order parameter. For a single-mode laser, spontaneous emission into non-cavity modes (including free space) provides a reservoir, and the thermodynamic limit is approached as the fraction of emission into this reservoir tends to unity. In the microlaser literature, this is equivalent to the β factor tending to zero; in our case the equivalent limit is where Γ_{\downarrow} tends to Γ_0 . In this thermodynamic limit, the derivative of the order parameter becomes discontinuous as a function of pump rate, indicative of a true second-order phase transition. We make use of the broader concept of condensation that the population of non-condensing modes (e.g. free space) saturate even as the total population increases, and that the condensing modes’ populations continue to increase linearly with total population. This

definition connects the Penrose-Onsager definition to the general criteria, e.g. as laid out in Ref. [8]. To avoid any confusion we have changed “*Despite only being strictly defined in thermal equilibrium as the macroscopic occupation of the ground state, we are assuming here a broader concept of condensation...*” to “*While BEC is only strictly defined in thermal equilibrium as the macroscopic occupation of the ground state, we are assuming here a broader concept of condensation...*”

Comment 5: “The values of g^2 never approach the grand-canonical value of 2 in Fig. 6. Why is this the case?”

Reply: We including the following discussion in the end of Sec. IIE to address the point raised by the Referee:

“Note that, the fluctuations described here are of a different origin than those arising from the grand-canonical nature of a photon BEC [14], which predicts $g^{(2)}(0) = 2$. In the latter, a steady-state is achieved by a detailed balance between cavity loss and c.w. pumping, with the fluctuations being related with the coupling between the photons and the molecular grand-canonical reservoir in conditions of thermal equilibrium. There is time-translation symmetry and number fluctuations are damped within a 2ns timescale [14]. In contrast, the $g^{(2)}$ structure we identify here reflects the propagation of the initial fluctuations associated with the spontaneous emission events that trigger the growth of the condensate pulse. The system never reaches a steady-state and all the dynamics are fundamentally transient. In principle, $g^{(2)}(t, t)$, which depends on both the individual pulse shapes and their formation jitter, can even be larger than 2.”

Comment 6: “In the introduction (first line of second page), the concept transient phase transition is put forward. It is not clear what is meant with this. Do the authors not refer to the transient dynamics after quenching through a phase transition instead of a “transient phase transition”, which seems to indicate a different type of phase transition.”

Reply: We define a *transient phase transition* ‘...as the evolution in configuration space after a jump across a phase transition in parameter space (a quench), where “phase transition” has its usual time-independent, thermodynamic meaning. This is distinct from the recently introduced concept of dynamical phase transitions [15, 16]’ This new sentence is now in the text (Section II.A), along with a specific description for our system, and we hope this clarifies the Referee’s query.

Comment 7: “In the conclusions, they make the connection to ‘phase transitions’ in different systems. This may a bit misleading since the system under consideration does not have a phase transition in a precise sense. Because only one mode is considered (it is argued that the excited states are not important) it is actually rather a zero-dimensional system, that does not show a phase transition: all quantities are analytic functions of parameters, except in the limit where the number of photons goes to infinity. The final outlook of the paper is actually about what happens in spatially extended systems. It is actually well known that spatially extended systems that undergo sudden condensation feature Kibble-Zurek type vortices (cf. Ref. [9,10]). Maybe the authors could be more precise about the open problems in spatially extended systems.”

Reply: In our reply above we have defined the thermodynamic limit for our system carefully. It is the accessibility of modes in free space which allows us to define this limit for our otherwise single-mode system. Regarding the dynamics of spatially extended systems, the Referee pertinently mentions the Kibble-Zurek mechanism, which is related to the type of physics we anticipate to have important connection with our results. In particular, we now further describe in the manuscript: “*In the context of the Kibble-Zurek mechanism [17, 18], for instance, most studies are simply concerned with the defect number scaling after the system relaxes to some steady-state, with the intrinsic relaxation dynamics often ignored. As such, although we cannot anticipate specific effects, one wonders about the correspondence between transient fluctuation dynamics of the zero-dimensional system described in our paper and that of spatially extended systems.*”

[1] E. Jaynes, *Frontiers of Nonequilibrium Statistical Physics* (Springer, New York, 1986), pp. 33–55.

[2] J. R. Tucker and B. I. Halperin, *Phys. Rev. B* **3**, 3768 (1971).

[3] P. C. Hohenberg and B. I. Halperin, *Rev. Mod. Phys.* **49**, 435 (1977).

[4] G. Coslovich, C. Giannetti, F. Cilento, S. Dal Conte, T. Abebaw, D. Bossini, G. Ferrini, H. Eisaki, M. Greven, A. Damascelli, and F. Parmigiani, *Phys. Rev. Lett.* **110**, 107003 (2013).

[5] L. Perfetti, B. Sciolla, G. Biroli, C. J. van der Beek, C. Piovera, M. Wolf, and T. Kampfrath, *Phys. Rev. Lett.* **114**, 067003 (2015).

[6] P. Kirton and J. Keeling, *Phys. Rev. Lett.* **111**, 100404 (2013).

- [7] J. Schmitt, T. Damm, D. Dung, C. Wahl, F. Vewinger, J. Klaers, and M. Weitz, *Phys. Rev. Lett.* **116**, 033604 (2016).
- [8] J. Knebel, M. F. Weber, T. Krüger, and E. Frey, *Nature Communications* **6**, 6977 EP (2015).
- [9] J. Polte, *CrystEngComm* **17**, 6809 (2015).
- [10] N. T. K. Thanh, N. Maclean, and S. Mahiddine, *Chemical Reviews* **114**, 7610 (2014).
- [11] A. Lebreton, I. Abram, R. Braive, I. Sagnes, I. Robert-Philip, and A. Beveratos, *Phys. Rev. Lett.* **110**, 163603 (2013).
- [12] A. Lebreton, I. Abram, R. Braive, N. Belabas, I. Sagnes, F. Marsili, V. B. Verma, S. W. Nam, T. Gerrits, I. Robert-Philip, M. J. Stevens, and A. Beveratos, *Applied Physics Letters* **106**, 031108 (2015).
- [13] G. Moody, M. Segnon, I. Sagnes, R. Braive, A. Beveratos, I. Robert-Philip, N. Belabas, F. Jahnke, K. L. Silverman, R. P. Mirin, M. J. Stevens, and C. Gies, *Optica* **5**, 395 (2018).
- [14] J. Schmitt, T. Damm, D. Dung, F. Vewinger, J. Klaers, and M. Weitz, *Phys. Rev. Lett.* **112**, 030401 (2014).
- [15] M. Heyl, A. Polkovnikov, and S. Kehrein, *Phys. Rev. Lett.* **110**, 135704 (2013).
- [16] M. Heyl, *Reports on Progress in Physics* **81**, 054001 (2018).
- [17] N. Navon, A. L. Gaunt, R. P. Smith, and Z. Hadzibabic, *Science* **347**, 167 (2015).
- [18] C. N. Weiler, T. W. Neely, D. R. Scherer, A. S. Bradley, M. J. Davis, and B. P. Anderson, *Nature* **455**, 948 EP (2008).

REVIEWERS' COMMENTS:

Reviewer #1 (Remarks to the Author):

With the new revisions the authors have significantly improved the clarity of their manuscript. This addresses most of my criticisms of the previous version. I think the general framework they present is interesting and the good agreement between experiments and theory is impressive. Hence, the manuscript should be published in Nature Communications. However, I still have one concern about the presentation of the free energy landscape:

As far as I can see the expression in Eqn 7 is an equilibrium result. It is derived in the limit where all of the non-equilibrium processes are not present and the number of excitations is constant. The fluctuations which give rise to the probability distributions in Fig 9 are non-trivial since they depend on n . I think it is interesting that such an approach can give a good understanding of the experimental results but the emphasis placed on the non-equilibrium aspects of the theory are still not fully clear to me.

There are also some minor points which the authors may like to consider

- 1) Would it be useful to add the variance of the trajectories to Fig 7? This would allow the early time jitter to be more clearly seen.
- 2) The notation in eqn 7 is slightly hard to read. Maybe it would be clearer to leave in the explicit time derivative instead of the dot?
- 3) I don't quite understand the relationship between the number distribution and the time jitter in Fig. 9. Can the authors be more clear as to why it is that when the n distribution is wide the time jitter is small.

Reviewer #2 (Remarks to the Author):

To my judgement, the authors have responded adequately to the referees remarks. I think the revised manuscript can be accepted for publication in Nature Communications.

Reviewer #3 (Remarks to the Author):

The authors have answered convincingly to my remarks and improved the clarity of the manuscript. I now recommend the publication of the manuscript in Nature Communications.

Reply to Referees

Title: Nonstationary Statistics and Formation Jitter in Transient Photon Condensation

Authors: João D. Rodrigues, and co-authors, Ben Walker, Himadri Dhar, Rupert Oulton, Florian Mintert and Rob Nyman

I. REPLY TO REFEREE 1

Main comment: “With the new revisions the authors have significantly improved the clarity of their manuscript. This addresses most of my criticisms of the previous version. I think the general framework they present is interesting and the good agreement between experiments and theory is impressive. Hence, the manuscript should be published in Nature Communications. However, I still have one concern about the presentation of the free energy landscape: As far as I can see the expression in Eqn 7 is an equilibrium result. It is derived in the limit where all of the non-equilibrium processes are not present and the number of excitations is constant. The fluctuations which give rise to the probability distributions in Fig 9 are non-trivial since they depend on n . I think it is interesting that such an approach can give a good understanding of the experimental results but the emphasis placed on the non-equilibrium aspects of the theory are still not fully clear to me.”

Reply: The effective free energy defined by Eq. (7) is obtained in the limit of no losses and pumping, thus describing a closed system which, naturally, always evolves towards equilibrium in the long time limit. Nonetheless, even a closed system can be prepared in a far-from-equilibrium configuration. While typical (equilibrium) free energy descriptions are generally valid only near the equilibrium configuration, the validity of our effective free energy extends to all possible configurations (all possible photon numbers) and correctly describes the relaxation of the photon condensate when initialized at any non-equilibrium configuration, within the closed system approximation. The initial incoherent pumping of the cavity corresponds to such a preparation of the system into a highly non-equilibrium configuration. In order to make this point more clear, we have slightly rewritten the paragraph that contains Eq. (7) (marked in red), which we invite the Referee and the Editor to examine.

Minor comment 1: “Would it be useful to add the variance of the trajectories to Fig 7? This would allow the early time jitter to be more clearly seen.”

Reply: Note that, the variance of the trajectories is already shown in Fig. (6), given the relation $g^{(2)}(t, t) \simeq 1 + \langle \Delta n(t)^2 \rangle / \langle n(t) \rangle^2$. Moreover, the parameters of Fig. (7) match those of Fig. (6).

Minor comment 2: “The notation in eqn 7 is slightly hard to read. Maybe it would be clearer to leave in the explicit time derivative instead of the dot?”

Reply: Modified accordingly. For the sake of consistency, the “dot” notation was modified throughout the rest of the manuscript.

Minor comment 3: “I don’t quite understand the relationship between the number distribution and the time jitter in Fig. 9. Can the authors be more clear as to why it is that when the n distribution is wide the time jitter is small.”

Reply: In order to further clarify this relation, we have modified Fig. (9), which now includes a panel depicting the PDF of the condensate formation time, obtained from the free energy model. The caption as well as part of the text were slightly edited (in red) to make this connection more clear.

II. REPLY TO REFEREE 2

Main comment: “To my judgement, the authors have responded adequately to the referees remarks. I think the revised manuscript can be accepted for publication in Nature Communications.”

Reply: We thank the Referee for the positive final report.

III. REPLY TO REFEREE 3

Main comment: “The authors have answered convincingly to my remarks and improved the clarity of the manuscript. I now recommend the publication of the manuscript in Nature Communications.”

Reply: We thank the Referee for the positive final report.